# Association between Emotional Intelligence and Stress Coping Strategies According to Sex in Mexican General Population

**DOI:** 10.3390/ijerph19127318

**Published:** 2022-06-14

**Authors:** Fabiola Macías-Espinoza, Aniel Jessica Leticia Brambila-Tapia, Yesica Arlae Reyes-Domínguez, María Luisa Ramírez-García

**Affiliations:** 1Departamento de Psicología Aplicada, Centro Universitario de Ciencias de la Salud (CUCS), Universidad de Guadalajara, Guadalajara 44340, Jalisco, Mexico; fabiola.macias@academicos.udg.mx; 2Departamento de Psicología Básica, Centro Universitario de Ciencias de la Salud (CUCS), Universidad de Guadalajara, Guadalajara 44340, Jalisco, Mexico; 3Maestría en Psicología de la Salud, Centro Universitario de Ciencias de la Salud (CUCS), Universidad de Guadalajara, Guadalajara 44340, Jalisco, Mexico; yesica.reyes2393@alumnos.udg.mx (Y.A.R.-D.); maria.ramirez4357@alumnos.udg.mx (M.L.R.-G.)

**Keywords:** emotional intelligence, coping styles, sex, cognitive active coping

## Abstract

Emotional intelligence has been associated with adaptive coping in the adolescent and young population; however, the association of specific dimensions of emotional intelligence with each coping strategy has not been associated in general nor by each sex separately. Therefore, the aim of the study was to determine such an association. The general population was invited to perform an electronic questionnaire via social networks. A sample of 984 individuals were included, from which 62.1% were women, in whom we detected higher levels of emotional attention, and lower levels of emotional clarity and emotional repair, as well as increased levels of stress, depression and anxiety than men. In the bivariate correlations we observed significant positive correlations between emotional attention with stress, depression and anxiety, and significant negative correlations between emotional clarity and emotional repair with the three negative psychological variables, in both sexes. Adaptive coping strategies (mainly active coping and planning) showed positive correlations with emotional attention, emotional clarity and repair, being higher for emotional clarity and repair in both sexes. In addition, these two subscales also showed low negative correlations between some maladaptive strategies in both sexes, which suggests that interventions addressed to increase these emotional abilities could be useful in increasing adaptive coping.

## 1. Introduction

Emotional intelligence is defined as the individuals’ beliefs about their own emotional abilities, including the ability to observe and think about their feelings as well as to understand and regulate their own emotional states [1]. On the other hand, coping is defined as: “individual’s attempts to use cognitive and behavioral strategies to manage and regulate pressures, demands and emotions in response to stress” [2]. Sex differences have been described between coping strategies and their relation to psychological and physiological variables [3,4,5,6]; likewise, sex differences have been described in emotional intelligence, with higher emotional attention and lower emotional clarity and repair in girls when compared with boys [7], where emotional clarity and repair were also related to adaptive responses to positive and negative affect. In addition, it has been shown that emotional intelligence, specifically emotion recognition and expression, emotion understanding and emotion management and control, was associated with increased problem-solving coping in adolescents, which, in turn, was negatively corelated with negative psychological and behavioral variables [8]. In line with these findings, an additional report has shown a relationship between pro-active coping strategies and emotional attention, emotional clarity and emotional repair in Chinese pilots, being higher for the subscales of emotional clarity and repair [9] and a previous report of our research group showed that emotional clarity and emotional repair correlated positively with treatment adherence in patient with rheumatoid arthritis, and that an intervention addressed to increase these emotional abilities increased treatment adherence in this disease [10]. Nevertheless, to the best of our knowledge, no study has documented the correlation between emotional intelligence and the specific coping strategies in the adult general population or analyzed it by sex. Therefore, we decided to perform such a study in order to detect possible relations between emotional abilities (emotional attention, clarity and repair) with specific coping styles in a bivariate correlation, and with cognitive active coping in multivariate correlation for each sex in a sample of the adult Mexican general population.

## 2. Subjects and Method

An electronic questionnaire with sociodemographic and psychological instruments was send via social networks including WhatsApp, Facebook and e-mail, to the general population by the research team; this population included: university students, familiars, friends, colleagues and acquaintances of the research team, many of whom re-sent the questionnaire. The study was approved by the ethics and research committee of the Health Sciences University Center of the University of Guadalajara and the participants gave their consent to participate in the same questionnaire.

The socio-demographic data included: sex, age, whether they have a romantic partner, schooling, whether they have children, whether they have a job, socio-economic level, monthly extra money (excluding necessary expenses), daily free hours and weekly physical activity hours.

The psychological measures included: stress, measured with the Cohen perceived stress scale (CPSS) [11,12], with the range of the instrument being 1–5; depression, measured with the CES-D scale [13,14], with the range of the instrument being 0–3; anxiety, measured with the GAD-7 scale [15,16], with the range of the instrument being 0–3, emotional intelligence, measured with the Trait Meta Mood Scale (TMMS-24) [1,17], with the range of the instrument being 1–5; and coping strategies, measured with the brief-COPE scale [18,19], with the range of the instrument being 0–3.

### Statistical Analysis

To describe qualitative variables, we used frequencies and percentages, and for quantitative variables, means and standard deviation were used. In order to compare the psychological variables between the sexes, the Mann–Whitney U test was used, which considers the non-parametric distribution of the variables, and for comparison between the categories of the variables, we used the chi-squared test and Fisher exact test. Cronbach’s alpha was used to determine the reliability of each scale and subscale utilized. To perform comparisons between the psychological variables, we used the Spearman correlation test, considering the non-parametric distribution of the variables. In order to detect the distribution of the data, the Kolmogorov–Smirnov test was used. Finally, a multiple regression analysis, with the stepwise method, for cognitive active coping (the sum of active coping and planning) by each sex, was performed in order to determine the variables most associated with this adaptive coping in men and women. In this analysis we only included the sociodemographic variables and the three subscales of emotional intelligence. All analyses were carried out with the software SPSS v. 25, and a *p* value < 0.05 was considered as significant.

## 3. Results

A total of 984 participants were included. Of these, 611 (62.1%) were women; the sociodemographic data of participants is shown in Table 1. We observed that both sexes were comparable in age, schooling, whether they have children and socioeconomic level. However, men showed significantly higher levels of daily free hours, weekly physical activity hours, smoking frequency and alcohol consumption frequency than women (*p* < 0.05).

The Cronbach’s alpha for the subscales of emotional intelligence was above 0.9. In the case of the subscales of the brief-COPE, most of them had scores above 0.6, with the exception of self-distraction, behavioral disengagement, denial and acceptance, which had scores above 0.5. In the case of the subscale venting, the Cronbach’s alpha was low at 0.35; therefore, we did not use this subscale to perform correlations.

### 3.1. Comparisons of Emotional Intelligence, Cognitive Active Coping and Psychological Variables between Sexes

Women showed lower levels of emotional clarity (with a borderline *p* value) and lower levels of emotional repair than men. In addition, women also showed higher levels of emotional attention than men. A significant decrease in cognitive active coping was observed in women with respect to men. These differences were observed when the variables are compared numerically or categorically (divided in low, medium and high levels) (Table 2). With respect to the negative psychological variables, women showed higher levels of depression, anxiety and stress than men (Table 2).

### 3.2. Correlations between Cognitive Active Coping and Sociodemographic Variables

In the bivariate correlations between cognitive active coping with sociodemographic variables in women, we observed that there was a positive correlation between cognitive active coping with monthly extra money (r = 0.216, *p* < 0.001) and a negative correlation with having children (r = −0.088, *p* = 0.028). In men, there were positive correlations between cognitive active coping with monthly extra money (r = 0.160, *p* = 0.002), schooling (r = 0.134, *p* = 0.009), having a romantic partner (r = 0.133, *p* = 0.009) and having children (r = 0.101, *p* = 0.048) (Table 3).

### 3.3. Correlations between Emotional Intelligence with Coping Strategies and Psychological Variables

In the bivariate correlations, we observed low positive correlations between emotional attention and stress, depression and anxiety; likewise, we detected low negative correlations between emotional clarity and repair with the three negative psychological variables in both sexes (Table 4).

In the correlation analysis between the emotional intelligence subscales (emotional attention, emotional clarity and emotional repair) with the adaptive coping strategies, we observed moderate positive correlations between active coping and planning with emotional clarity and repair in both sexes; for these strategies, emotional attention also showed low positive correlations.

In addition, the coping strategies, acceptance, positive reframing, emotional support, instrumental support and religion, also showed low positive correlations with the three subscales of emotional intelligence in both sexes. These correlations were moderate between emotional repair and positive reframing in both sexes.

In the case of maladaptive coping styles, we observed low negative correlations between self-blame, and emotional clarity and repair in women and low positive correlations between self-blame and emotional attention in both sexes. Behavioral disengagement also showed low negative correlations with emotional clarity and repair in both sexes. In addition, denial showed a low negative correlation with emotional clarity and repair in women, and with emotional clarity in men. Self-distraction, showed low positive correlations with emotional attention and emotional repair in both sexes. Finally, substance use showed low negative correlations with emotional clarity and repair in men, and with emotional clarity in women (Table 4).

### 3.4. Multivariate Regression Analysis for Cognitive Active Coping

In the multivariate regression analysis for cognitive active coping (the sum of active coping and planning) in both sexes, we showed that emotional repair followed by emotional clarity were the most associated variables with this coping style; in the case of women, emotional attention also contributed in a lesser extent. In women, the sociodemographic variables of monthly extra money (positively associated) and having children (negatively associated) were also included in the model; while in men, schooling (positively associated) and daily free hours (negatively associated) were also included in the model (Table 5).

## 4. Discussion

In this study, we observed that the three subscales of emotional intelligence showed differences between the sexes. Emotional attention was significantly higher in women than in men, while emotional clarity and emotional repair were lower in women than in men (reaching significance only in emotional repair). Likewise, we observed that cognitive active coping was significantly lower in women than in men and that the three negative psychological variables measured (anxiety, stress and depression) were significantly higher in women when compared with men. In addition, significant positive correlations between cognitive active coping with the three subscales of emotional intelligence were found in both sexes, being higher for emotional clarity and emotional repair.

We observed that, similar to the study of Gómez-Baya and Mendoza (2018) [7], women showed higher levels of emotional attention and lower levels of emotional clarity and emotional repair when compared with men. In addition, we observed that emotional attention was positively corelated with stress, anxiety and depression, and emotional clarity and emotional repair were negatively correlated with these variables in both sexes. The low positive association between emotional attention and stress, anxiety, and depression in both sexes can be explained when considering emotional attention being a consequence rather than a cause of higher levels of these negative psychological variables. Globally, these observations, together with the significant lower levels of cognitive active coping in women when compared with men, can explain the higher levels of stress, depression and anxiety in women than in men.

In addition, emotional clarity and emotional repair were positively correlated with all subscales of adaptive coping, mainly with the cognitive active coping subscales (active coping and planning), which was confirmed in the multiple regression analysis in both sexes, with these variables being the most associated with this adaptive coping style. These observations coincide with previous reports showing a positive association between emotional intelligence and adaptive coping styles in adolescents [7,8] and with the study of Guo et al. (2017), who reported positive correlations between the three subscales of emotional intelligence with pro-active coping in Chinese pilots [9]. These results are also in line with previous reports showing that interventions based on emotional intelligence showed significant improvements in adaptive coping styles and treatment adherence, a desirable behavior in people with chronic conditions [10,20]; however, the associations performed with these coping styles in the adult population and analyzed by gender had not been reported so far.

The relationship between emotional intelligence, specifically emotional clarity and repair, with adaptive coping styles, can be explained by considering that the ability to identify and modify the emotions could contribute to the better comprehension and correct solving of stressful situations, probably because these abilities can improve the cognitive function, in addition to improving psychological wellbeing, which permits the individual to appreciate more options to face the problem.

We also observed negative correlations between some maladaptive coping styles (behavioral disengagement, denial and substance use) and emotional clarity and/or emotional repair in both sexes. Self-blame also showed negative correlations between emotional clarity and emotional repair only in women. These results suggest that emotional clarity and emotional repair are emotional abilities that are useful in order to increase adaptive coping styles in both sexes and prevent maladaptive coping styles, mainly in women. However, considering that the correlations with adaptive coping styles were higher than those with maladaptive coping styles, the increase in emotional clarity and repair would be more useful in increasing adaptive coping styles (mainly active coping styles) than in decreasing maladaptive ones.

In the multivariate regression analysis performed for the cognitive active coping style, we observed that emotional attention was included in the model for women but not for men, which coincides with the higher correlation between emotional attention with active coping and planning in women than in men, and could indicate that emotional attention is more important for active coping in women than in men. In addition, different sociodemographic variables were included in the regression model for each sex, with a low association of monthly extra money in women and of schooling in men, which coincides with the bivariate correlations. However, in the bivariate correlations we observed that monthly extra money was positively correlated with this coping style in both sexes, which is explained by considering that the active resolution of problems permits more economic solvency. In the case of men, the cognitive active coping also correlated with schooling, having a romantic partner and having children, which suggests that this adaptive style may facilitate these living conditions.

The study has the following limitations: The sample is predominantly young and was not randomly selected, so the representativeness all of the Mexican population can be diminished and restricted to young and educated people, under-representing the older people and those from lower socioeconomic levels. In addition, the cross-sectional design of the study does not permit us to demonstrate causality between the studied variables, being plausible bilateral relationships between them. In this sense, it would be also possible that increased stressful conditions could also increase maladaptive coping strategies and diminish adaptive ones. Finally, the menstrual cycle of the participants can be a confusing variable that was not measured, with it being possible that the specific menstrual cycle of women of reproductive age could also be related with the studied psychological variables; however, a recent report showed that maladaptive strategies were associated with depression, adjusting for the menstrual cycle, among stressed women struggling to conceive [21].

## 5. Conclusions

The subscales of emotional intelligence, emotional clarity and emotional repair, were lower in women than in men and were negatively associated with stress, depression and anxiety in both sexes. In addition, they were positively associated with adaptive coping styles in both sexes, with a higher correlation with cognitive active coping. These subscales of emotional intelligence were also negatively correlated with some maladaptive coping styles in both sexes, which suggests that the increase in these emotional abilities could increase adaptive coping styles and decrease maladaptive ones, and additionally decrease stress, depression and anxiety.

## Figures and Tables

**Table 1 ijerph-19-07318-t001:** Sociodemographic variables in the studied population.

Variable	Women, *n* = 611	Men, *n* = 373
Age, mean ± SD	30.19 ± 11.02	30.38 ± 11.32
With a romantic partner, *n* (%)	378 (61.9)	204 (54.7)
With children, *n* (%)	192 (31.4)	102 (27.3)
With a job, *n* (%)	373 (61.0)	258 (69.2)
Educational level		
- Elementary school	1 (0.20)	0 (0.0)
- High school	10 (1.60)	7 (1.9)
- Preparatory	125 (20.5)	86 (23.0)
- Bachelor’s degree	349 (57.1)	204 (54.7)
- Technical career	29 (4.7)	20 (5.4)
- Master degree	74 (12.1)	36 (9.6)
- Ph.D. degree	23 (3.8)	20 (5.4)
Socioeconomic level		
- Very low	0 (0.0)	4 (1.1)
- Low	102 (16.7)	60 (16.1)
- Medium	490 (80.2)	301 (80.7)
- High	19 (3.1)	8 (2.1)
Daily free hours, mean ± SD	4.02 ± 2.64	4.52 ± 2.77
Weekly physical activity hours, median (range)	2 (0–20)	3 (0–35)
Smoking frequency, mean ± SD	1.62 ± 1.42	1.99 ± 1.75
Alcohol consumption, mean ± SD	2.70 ± 1.39	3.11 ± 1.54

SD: Standard deviation. Smoking and alcohol consumption were measured from 1: never to 6: many times, in the week.

**Table 2 ijerph-19-07318-t002:** Comparison of emotional intelligence and negative psychological variables between sexes.

Variable	Men (*n* = 373)	Women (*n* = 611)	*p* Value
**Emotional clarity, Mean (SD)**	3.23 (1.05)	3.10 (1.02)	0.083
**Categorized *n* (%)**			
-** Low**	22 (5.9)	44 (7.2)	0.329
-** Medium**	181 (48.5)	317 (51.9)	
-** High**	170 (45.6)	250 (40.9)	
**Emotional repair, Mean ± SD**	3.31 ± 0.95	3.16 ± 0.95	0.013
**Categorized *n* (%)**			
-** Low**	14 (3.7)	24 (3.9)	0.049
-** Medium**	173 (46.4)	331 (54.2)	
-** High**	186 (49.9)	256 (41.9)	
**Emotional attention, Mean ± SD**	2.94 ± 0.99	3.18 ± 0.96	<0.001
**Categorized *n* (%)**			
-** Low**	30 (8.0)	34 (5.6)	<0.001
-** Medium**	216 (58.0)	319 (52.2)	
-** High**	127 (34.0)	258 (42.2)	
**Cognitive active coping**	1.91 ± 0.66	1.80 ± 0.68	0.004
**Categorized *n* (%)**			
-** Low**	77 (20.6)	188 (30.8)	0.002
-** Medium**	238 (63.9)	336 (55.0)	
-** High**	58 (15.5)	87 (14.2)	
**Negative psychological variables**			
**Stress, Mean ± SD**	2.71 ± 0.67	3.02 ± 0.66	<0.001
**Categorized *n* (%)**			
-** Low**	23 (6.2)	15 (2.5)	<0.001
-** Medium**	291 (78.0)	409 (66.9)	
-** High**	59 (15.8)	187 (30.6)	
**Depression, Mean ± SD**	1.02 ± 0.61	1.27 ± 0.65	<0.001
**Categorized *n* (%)**			
-** Low**	272 (72.9)	351 (57.4)	<0.001
-** Medium**	97 (26.0)	252 (41.3)	
-** High**	4 (1.1)	8 (1.3)	
**Anxiety, Mean ± SD**	1.01 ± 0.75	1.35 ± 0.81	<0.001
**Categorized *n* (%)**			
-** Low**	270 (72.4)	344 (56.3)	<0.001
-** Medium**	91 (24.4)	216 (35.4)	
-** High**	12 (3.2)	51 (8.3)	

Emotional intelligence subscales and stress scale (CPSS) had a range of 1–5; depression scale (CES-D) range: 0–3; anxiety scale (GAD-7) range: 0–3; cognitive active coping (brief-COPE), range: 0–3.

**Table 3 ijerph-19-07318-t003:** Correlations between cognitive active coping with the sociodemographic variables.

Variable	Women (*n* = 611)	Men (*n* = 373)
Age	0.041	0.081
Schooling	0.070	0.134 **
Having children	−0.088 *	0.102 *
With a romantic partner	0.049	0.133 **
With a job	0.043	0.073
Socioeconomic level	0.061	0.052
Monthly extra money	0.216 **	0.160 **
Weekly physical activity hours	0.058	0.090
Daily free hours	0.026	−0.092
Smoking frequency	−0.035	−0.002
Alcohol consumption frequency.	−0.032	−0.024

* *p* < 0.05, ** *p* < 0.01. *p* value obtained with Spearman correlation test.

**Table 4 ijerph-19-07318-t004:** Correlations between emotional intelligence and coping strategies in each sex.

	Women (*n* = 611)			Men (*n* = 373)		
Variable	Emotional Attention	Emotional Clarity	Emotional Repair	Emotional Attention	Emotional Clarity	Emotional Repair
**Stress**	0.119 **	−0.345 **	−0.343 **	0.103 *	−0.372 **	−0.304 **
**Depression**	0.132 **	−0.315 **	−0.343 **	0.110 *	−0.290 **	−0.279 **
**Anxiety**	0.245 **	−0.163 **	−0.212 **	0.130 *	−0.221 **	−0.155 **
**Coping strategies**						
**Acceptance**	0.217 **	0.339 **	0.386 **	0.224 **	0.368 **	0.325 **
**Humor**	0.245 **	0.094 *	0.224 **	0.132 **	0.154 **	0.282 **
**Religion**	0.075	0.206 **	0.292 **	0.129 *	0.169 **	0.213 **
**Substance use**	0.056	−0.137 **	−0.071	0.050	−0.123 *	−0.125 *
**Self-blame**	0.216 **	−0.168 **	−0.153 **	0.242 **	−0.085	0.033
**Behavioral disengagement**	0.042	−0.197 **	−0.157 **	0.018	−0.227 **	−0.166 **
**Emotional support**	0.382 **	0.272 **	0.264 **	0.287 **	0.146 **	0.125 *
**Instrumental support**	0.318 **	0.145 **	0.146 **	0.319 **	0.202 **	0.219 **
**Active coping**	0.306 **	0.413 **	0.404 **	0.282 **	0.499 **	0.490 **
**Planning**	0.272 **	0.413 **	0.439 **	0.193 **	0.426 **	0.427 **
**Self-distraction**	0.191 **	0.047	0.187 **	0.109 *	−0.040	0.171 **
**Denial**	0.077	−0.186 **	−0.132 **	0.102 *	−0.137 **	−0.031
**Positive reframing**	0.188 **	0.281 **	0.518 **	0.242 **	0.304 **	0.474 **
**Cognitive active coping**	0.314 **	0.450 **	0.458 **	0.263 **	0.506 **	0.501 **

* *p* < 0.05, ** *p* < 0.01. *p* value obtained with Spearman correlation test.

**Table 5 ijerph-19-07318-t005:** Multivariate regression analysis for cognitive active coping in women and men.

Women				
Variable	Beta	Beta Coefficient	*p* Value	Change in R^2^
**Constant**	0.718	-	0.001	-
**Emotional repair**	0.391	0.272	0.000	0.190
**Emotional clarity**	0.251	0.187	0.000	0.041
**Monthly extra money**	0.225	0.186	0.000	0.028
**Having children**	−0.344	0.116	0.001	0.017
**Emotional attention**	0.137	0.097	0.021	0.007
**Men**				
**Constant**	0.629	-	0.060	-
**Emotional repair**	0.461	0.334	0.000	0.284
**Emotional clarity**	0.406	0.326	0.000	0.071
**Schooling**	0.109	0.089	0.034	0.008
**Daily free hours**	−0.041	−0.087	0.037	0.008

R of the model for women: 0.532, R of the model for men: 0.609.

## Data Availability

Data that support the findings of the study are available from the corresponding author upon reasonable request.

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
