# Peer review of "Association between Emotional Intelligence and Stress Coping Strategies According to Sex in Mexican General Population"

_ijerph, 2022, doi:10.3390/ijerph19127318_

Round 1
Reviewer 1 Report
To date, the dependence of the cognitive and affective functions of a person on the level of hormones has been proven (for review Abo S, Smith D, Stadt M, Layton A. Modelling female physiology from head to Toe: Impact of sex hormones, menstrual cycle, and pregnancy. J Theor Biol. 2022 May 7;540:111074. doi: 10.1016/j.jtbi.2022.111074.. It is also well known that the intensity of experiencing emotions, stress, of emotional attention, depression. and other cognitive and emotional functions change dramatically during a woman's menstrual cycle, which is also associated with different levels of hormones at different stages of the menstrual cycle (more than 1500 origins in PubMed https://pubmed.ncbi.nlm.nih.gov/?term=menstrual+cycle+emotions&sort=date).
Studies of differences between men and women that do not take into account the activity of hormones (the phase of the female menstrual cycle) are puzzling. The question naturally arises: was the level of sex steroids the same in the compared groups? Unlikely! Therefore, women cannot be considered as a homogeneous group, since systemic physiological, neuronal and psychological parameters change in the same woman during the month depending on the hormonal phase. In the same woman, different emotional and cognitive statuses coexist in different periods.
The huge work done loses its meaning without taking into account the above and is not completed. Meanwhile, there are enough participants in the experiment to take into account at least three hormone-dependent periods: low levels of estrogen and progesterone (menstrual and premenstrual phases), an increase in estrogen (late follicular and ovulatory) and an increase in progesterone. (mid-luteal phase)
I suggest this research needs being completed with female participants menstrual cycle data. Sure, sex difference could be more valid due comparison of men with 3 groups of women in different hormonal conditions, excluding pregnant women and women hormonal contraceptive users.
Author Response
Response: We agree that the menstrual cycle can modify the mood, and therefore the variables studied (emotional intelligence, depression, anxiety, stress and coping strategies); however, the precise measurement of this information is difficult, this considering that many women do not know this information precisely, and its measurement without precision could affect the results.
Additionally, we found a previous report showing that maladaptive coping strategies were associated with depression among distressed women struggling to conceive, adjusting for the menstrual cycle, supporting the findings of this study. We added this reference in the limitations section.
On the other hand, it would be impossible to collect that data, retrospectively, because, although many participants wrote their e-mail, the correct measurement of that information, many months later, would be difficult and many participants would not answer the data.
Therefore, although the menstrual cycle was not measured in this study, we think that the data is reliable and relevant, so, in order to consider this confounding variable, we added additional information in the limitations of the study explaining this point.
Reviewer 2 Report
The idea of the authors' research looks interesting. They examined the associations between emotional abilities (emotional attention, clarity, and recovery) with specific coping styles in a bivariate correlation and with cognitive-active coping in a multivariate correlation for each gender. However, the implementation of this idea in the form of the presented article is far from optimal.
Major comments:
- It is noteworthy that the article is dominated by references to articles more than 10 years old. Only one link to the 2018 article. Unwittingly there are doubts about the topicality of the study. However, in recent years there have been enough publications on the subject of the Trait Meta-Mood Scale. An example of this is a recent publication by authors from Mexico (Self-Perceived Emotional Intelligence Levels in Nursing Students in Times of a Pandemic: Multivariate Representation; doi: 10.3390/ijerph19031811) published in the recent issue of Int J Environ Res Public Health (2022 Feb; 19 (3): 1811). It would also be useful to use in the manuscript data on Emotional Intelligence from a systematic review published in 2021 (Emotional Intelligence Measures: A Systematic Review; doi: 10.3390/healthcare9121696). There are many more such examples, so the authors should conduct a more thorough literature search.
- The Subjects and method section is written very succinctly, it needs to be presented more fully. As an example, authors can use the already mentioned article (doi: 10.3390/ijerph19031811).
- Among the individuals included in the study, there is a wide range in age - from 18 to 77 years, with an average age of 30 years. It is quite possible that for people of different ages in such a heterogeneous sample, the relationship between the indicators studied by the authors is different. However, the authors did not study the influence of the age of the subjects and these relationships.
- Also, the age range of the subjects (up to 77 years) raises doubts that the present manuscript corresponds to the subject of the special issue (Wellbeing and Mental Health among Students and Young People).
- There are clearly few tables in the submitted manuscript. The information about socio-demographic data in groups, and information about the bivariate correlations between cognitive-active coping with sociodemographic variables is presented in the text, it is better to present them in a tables. Table 3 is simply missing in the text of the manuscript.
- In the results section, the data of Multivariate regression analysis for cognitive-active coping are presented briefly, without corresponding numerical indicators. Since Table 3 is also missing, it is completely unclear what results were obtained by the authors and how adequately they are interpreted.
- I also think that in the sections Introduction and Discussion, the question of the relationship between emotional intelligence and coping strategies is not sufficiently disclosed, although there are recent publications on these keywords in addition to the publications mentioned by the authors (Gómez-Baya and Mendoza, 2018; Downey et al., 2010). The following can be cited as examples: doi: 10.1016/j.nedt.2017.11.013, doi: 10.3357/AMHP.4799.2017, doi: 10.1016/j.apnr.2017.03.001, and doi: 10.1017/sjp.2016.8
- In the Discussion is missing a Study Limitations section.
Author Response
Major comments:
- It is noteworthy that the article is dominated by references to articles more than 10 years old. Only one link to the 2018 article. Unwittingly there are doubts about the topicality of the study. However, in recent years there have been enough publications on the subject of the Trait Meta-Mood Scale. An example of this is a recent publication by authors from Mexico (Self-Perceived Emotional Intelligence Levels in Nursing Students in Times of a Pandemic: Multivariate Representation; doi: 10.3390/ijerph19031811) published in the recent issue of Int J Environ Res Public Health (2022 Feb; 19 (3): 1811). It would also be useful to use in the manuscript data on Emotional Intelligence from a systematic review published in 2021 (Emotional Intelligence Measures: A Systematic Review; doi: 10.3390/healthcare9121696). There are many more such examples, so the authors should conduct a more thorough literature search.
Response: We added some of these references in order to increase the literature review and enrich the introduction and discussion sections.
- The Subjects and method section is written very succinctly, it needs to be presented more fully. As an example, authors can use the already mentioned article (doi: 10.3390/ijerph19031811).
Response: We added more information to better describe the methods section.
- Among the individuals included in the study, there is a wide range in age - from 18 to 77 years, with an average age of 30 years. It is quite possible that for people of different ages in such a heterogeneous sample, the relationship between the indicators studied by the authors is different. However, the authors did not study the influence of the age of the subjects and these relationships.
Response: We added the correlations with sociodemographic data including the age in Table 3, and no correlations between age and cognitive-active coping were observed. In addition, age was considered in the multivariate regression analysis.
- Also, the age range of the subjects (up to 77 years) raises doubts that the present manuscript corresponds to the subject of the special issue (Wellbeing and Mental Health among Students and Young People).
Response: Although the rage of the age is wide, the mean and SD of it correspond to young people in both sexes.
- There are clearly few tables in the submitted manuscript. The information about socio-demographic data in groups, and information about the bivariate correlations between cognitive-active coping with sociodemographic variables is presented in the text, it is better to present them in a tables. Table 3 is simply missing in the text of the manuscript.
Response: We added the socio-demographic data in Table 1 and the bivariate correlations with sociodemographic variables in table 3. The missing table is Table 5.
- In the results section, the data of Multivariate regression analysis for cognitive-active coping are presented briefly, without corresponding numerical indicators. Since Table 3 is also missing, it is completely unclear what results were obtained by the authors and how adequately they are interpreted.
Response: We verified that all the tables are in the manuscript, so the information of the multivariate regression is clearer now with table 5.
- I also think that in the sections Introduction and Discussion, the question of the relationship between emotional intelligence and coping strategies is not sufficiently disclosed, although there are recent publications on these keywords in addition to the publications mentioned by the authors (Gómez-Baya and Mendoza, 2018; Downey et al., 2010). The following can be cited as examples: doi: 10.1016/j.nedt.2017.11.013, doi: 10.3357/AMHP.4799.2017, doi: 10.1016/j.apnr.2017.03.001, and doi: 10.1017/sjp.2016.8
Response: We added new information to these sections
- In the Discussion is missing a Study Limitations section.
Response: We added this information in the discussion.
Reviewer 3 Report
The study in this manuscript was conducted with an aim to unravel the association between emotional intelligence and stress coping strategies as per sex in Mexican General Population. The study was performed in 1009 subjects, out of which 611 were women and the findings suggest that emotional clarity and emotional repair were lower in women than in men and were negatively associated with stress, depression and anxiety in both the sexes. The findings are interesting; however, the following clarifications are required.
1) As per methodology section, the study was conducted on 1009 individuals and 611 were women, then women should have been 398, but both the tables 1 and 2 in result section show women were 373. How to reconcile this while discussing the results in totality.
2) The result section show that out of 1009, 692 (70.30%) did not have children and 632 (64.20%) had a job. The authors need to give break-up of men and women, out of any figure/variable, they mention.
3) In table 1, the values for emotional attention parameter are 2.94±0.99 for men and 3.18±0.96 for women, which do not seem to be significantly different as the standard deviation is high and the difference in the value of means is very less. This needs to be checked again.
4) Table 2 should be shown separately from table 3 and further the values of coefficient of regression be given in table 3.
Author Response
Reviewer 3:
The study in this manuscript was conducted with an aim to unravel the association between emotional intelligence and stress coping strategies as per sex in Mexican General Population. The study was performed in 1009 subjects, out of which 611 were women and the findings suggest that emotional clarity and emotional repair were lower in women than in men and were negatively associated with stress, depression and anxiety in both the sexes. The findings are interesting; however, the following clarifications are required.
- As per methodology section, the study was conducted on 1009 individuals and 611 were women, then women should have been 398, but both the tables 1 and 2 in result section show women were 373. How to reconcile this while discussing the results in totality.
Response: We corrected the total number of participants, so the numbers now coincide
- The result section show that out of 1009, 692 (70.30%) did not have children and 632 (64.20%) had a job. The authors need to give break-up of men and women, out of any figure/variable, they mention.
Response: We separated the information of men and women in table 1.
- In table 1, the values for emotional attention parameter are 2.94±0.99 for men and 3.18±0.96 for women, which do not seem to be significantly different as the standard deviation is high and the difference in the value of means is very less. This needs to be checked again.
Response: We verified the information and it is correct; however, in order to clarify it, we added the information of the psychological variables categorized in low, high and medium in table 2 and performed the chi-squared test in order to compare the frequencies in both sexes
4) Table 2 should be shown separately from table 3 and further the values of coefficient of regression be given in table 3.
Response: The information of multivariate regression (before in table 3) has been mistakenly missing, however now it is added in Table 5.
Reviewer 4 Report
The authors looked into the factors that are most linked to cognitive-active coping in both men and women (including subscales of active coping and planning). They imply that interventions to improve these emotional qualities could help both men and women develop adaptive coping. Their concept appears to be novel, and it has been done extremely well from a technical standpoint, making it worthy of publication. The abstract is well-written, the subjects are meaningful and insightful, and they are both valuable and relevant for future research. The paper is well-written. The document, however, requires proofreading because it contains multiple grammatical errors.
Author Response
Reviewer 4
The authors looked into the factors that are most linked to cognitive-active coping in both men and women (including subscales of active coping and planning). They imply that interventions to improve these emotional qualities could help both men and women develop adaptive coping. Their concept appears to be novel, and it has been done extremely well from a technical standpoint, making it worthy of publication. The abstract is well-written, the subjects are meaningful and insightful, and they are both valuable and relevant for future research. The paper is well-written. The document, however, requires proofreading because it contains multiple grammatical errors.
Response: We performed an English grammar review and correction.
Round 2
Reviewer 1 Report
I'm satisfied with the author's answers
Reviewer 2 Report
The authors significantly improved the manuscript: they slightly updated the list of references, added tables, and a section on study limitations. I have no significant comments on the text of the manuscript. There are doubts about the possibility of referring the manuscript to a Special Issue - "Wellbeing and Mental Health among Students and Young People" - this is at the discretion of the editor.